# A cross-sectional study of cardiovascular disease risk clustering at different socio-geographic levels in India

Anne C. Bischops [1,2 ✉], Jan-Walter De Neve [1], Ashish Awasthi[3], Sebastian Vollmer [4], Till Bärnighausen[1,5,6] & Pascal Geldsetzer [1,7]

Despite its importance for the targeting of interventions, little is known about the degree to which cardiovascular disease (CVD) risk factors cluster within different socio-geographic levels in South Asia. Using two jointly nationally representative household surveys, which sampled 1,082,100 adults across India, we compute the intra-cluster correlation coefficients (ICCs) of five major CVD risk factors (raised blood glucose, raised blood pressure, smoking, overweight, and obesity) at the household, community, district, and state level. Here we show that except for smoking, the level of clustering is generally highest for households, followed by communities, districts, and then states. On average, more economically developed districts have a higher household ICC in rural areas. These findings provide critical information for sample size calculations of cluster-randomized trials and household surveys, and inform the targeting of policies and prevention programming aimed at reducing CVD in India.

[1] Heidelberg Institute of Global Health, Medical Faculty and University Hospital, Heidelberg University, Heidelberg, Germany. [2] Department of General Pediatrics, Neonatology, and Pediatric Cardiology, University Children's Hospital, Duesseldorf, Germany. [3] Public Health Foundation of India, New Delhi, India. [4] Department of Economics & Centre for Modern Indian Studies, University of Goettingen, Göttingen, Germany. [5] Department of Global Health and Population, Harvard T.H. Chan School of Public Health, Boston, MA, USA. [6] Africa Health Research Institute, Somkhele, KwaZulu-Natal, South Africa. [7] Division of Primary Care and Population Health, Department of Medicine, Stanford University, Stanford, CA, USA. ✉email: anne-christine.bischops@uni-heidelberg.de

Cardiovascular disease (CVD) is the leading cause of mortality and disability-adjusted life years in India[1,2]. The prevalence of important CVD risk factors in the country, including diabetes, hypertension, and overweight, is high and projected to rise rapidly over the coming decades[3–7]. We recently estimated a national diabetes prevalence of 8% and a hypertension prevalence of 27% among adults in India in 2012-2014[8]. In addition, approximately every fifth person aged between 15 and 49 years in the country was estimated to be overweight in 2016[9]. Another preventable, yet highly prevalent, CVD risk factor in India is tobacco use. In 2017, an estimated 11% of adults in India aged 15 years or older were current smokers[10].

Slowing the progress of the CVD epidemic in India will require identifying and implementing cost-effective strategies for the prevention and treatment of CVD risk factors. Many of the factors that increase one's probability of developing CVD risk factors—including diet[11], characteristics of the built environment[12], social networks[13], and genetics[14]—tend to be shared to at least some degree by individuals in the same households and communities. It is, therefore, unsurprising that studies have shown that diabetes, hypertension, obesity, and smoking tend to co-occur in households and larger community structures, such as neighborhoods[15,16]. However, although there is a considerable body of evidence on this phenomenon from high-income settings, there is little evidence from low- and middle-income countries (LMICs), including India, on the degree of such clustering of CVD risk factors.

In this study, we use nationally representative household survey data from India to determine the intracluster correlation coefficients (ICCs) of five major CVD risk factors at each of four different socio-geographic levels (household, community, district, and state). In addition, we aim to ascertain how the degree of clustering of CVD risk factors varies between states and by household wealth. The motivation for this study is not to inform individual-level clinical management. Instead, our objective is to provide critical information for sample size calculations of cluster-randomized trials and household surveys. In addition, understanding the degree to which important CVD risk factors tend to co-occur within these socio-geographic levels is crucial to inform the targeting of appropriate interventions. For instance, policymakers need to decide whether to target a screening program for diabetes and hypertension at specific communities or types of households, or if they should instead disregard these socio-geographic units, such as by screening everyone above a certain age threshold.

## Results

**Sample characteristics.** Out of 1,618,359 adults in our final dataset, 1,103,476 (68.1%) participants had non-missing values for all CVD risk factors (BG, BP, smoking status, height, and weight) and were included in the analysis. Because socio-demographic information in the DLHS-4 and AHS was collected for all household members from the household head, participants with missing outcome data included eligible household members who were not present at the time of the study team visit. The DLHS-4 and AHS jointly contained data on 515,689 households, 17,841 communities, and 561 districts. 52.5% of participants were female and 42.0% were younger than 36 years (Table 1). 9.1% of participants were obese (BMI ≥ 27.5 kgm$^{-2}$), 7.7% had a raised BG, and 26.9% had a raised BP. Smoking was far more common among men (23.3%) than women (2.3%). In all, 39.0% of participants had received no formal schooling, 32.6% lived in an urban area, and 76.2% were currently married.

**Cluster characteristics.** The mean cluster size of the 515,689 included households was 3.4 participants; the median cluster size

was three participants (Table 2). In the 17,841 communities, a cluster consisted on average of 109.4 participants, with the median cluster size being 66 participants. On the district level, our analysis included 561 districts with a mean and median size of 2415 and 2270 participants, respectively. A state consisted of a mean and median of 57,427 and 59,792 participants, respectively. Cluster characteristics computed separately for each state can be found in the appendix (Supplementary Table 20).

**Clustering at different socio-geographic levels.** The ICCs ranged from 0.023 (95% CI, 0.012–0.036) for raised BP at the state level to 0.236 (95% CI, 0.234–0.239) for overweight at the household level (Table 3). Among the CVD risk factors, the ICC was highest for overweight at all socio-geographic levels except at the state level where smoking had the highest ICC. With the exception of smoking, the ICC increased as the socio-geographic units

### Table 1 Sample characteristics[a,b].

| Characteristic | Total | Male | Female |
|---|---|---|---|
| n | 1,103,476 | 524,525 (47.5) | 578,951 (52.5) |
| *Age group, n (%)* | | | |
| 18–25 years | 193,689 (17.6) | 94,804 (18.1) | 98,885 (17.1) |
| 26–35 years | 268,797 (24.4) | 119,950 (22.9) | 148,847 (25.7) |
| 36–45 years | 243,217 (22.0) | 111,741 (21.3) | 131,476 (22.7) |
| 46–55 years | 183,829 (16.7) | 86,591 (16.5) | 97,238 (16.8) |
| 56–65 years | 129,033 (11.7) | 65,211 (12.4) | 63,822 (11.0) |
| >65 years | 84,896 (7.7) | 46,220 (8.8) | 38,676 (6.7) |
| Missing (%) | 0.0 | 0.0 | 0.0 |
| *BMI group, n (%)* | | | |
| >16.0 kgm$^{-2}$ | 42,473 (3.8) | 16,029 (3.1) | 26,444 (4.6) |
| 16.0–18.4 kgm$^{-2}$ | 164,509 (14.9) | 75,411 (14.4) | 89,098 (15.4) |
| 18.5–22.9 kgm$^{-2}$ | 521,591 (47.3) | 257,281 (49.1) | 264,310 (45.7) |
| 23.0–24.9 kgm$^{-2}$ | 165,114 (15.0) | 84,085 (16.0) | 81,029 (14.0) |
| 25.0–27.4 kgm$^{-2}$ | 109,266 (9.9) | 51,914 (9.9) | 57,352 (9.9) |
| 27.5–29.9 kgm$^{-2}$ | 52,797 (4.8) | 22,335 (4.3) | 30,462 (5.3) |
| ≥30.0 kgm$^{-2}$ | 47,726 (4.3) | 17,470 (3.3) | 30,256 (5.2) |
| Current smoker, n (%) | 135,736 (12.3) | 122,459 (23.3) | 13,277 (2.3) |
| Raised BG, n (%) | 85,327 (7.7) | 41,524 (7.9) | 43,803 (7.6) |
| Raised BP, n (%) | 296,634 (26.9) | 154,952 (29.5) | 141,682 (24.5) |
| *Education, n (%)* | | | |
| <Primary school | 428,392 (39.0) | 152,690 (29.2) | 275,702 (47.8) |
| Primary school | 136,620 (12.4) | 68,198 (13.1) | 68,422 (11.9) |
| Middle school | 165,682 (15.1) | 88,388 (16.9) | 77,294 (13.4) |
| Secondary school | 151,941 (13.8) | 85,326 (16.3) | 66,615 (11.5) |
| High school | 106,984 (9.7) | 61,185 (11.7) | 45,799 (7.9) |
| >High school | 109,567 (10.0) | 66,415 (12.7) | 43,152 (7.5) |
| Missing (%) | 0.4 | 0.4 | 0.3 |
| *Household wealth quintile, n (%)* | | | |
| 1 (least wealthy) | 229,326 (20.8) | 108,150 (20.6) | 121,176 (20.9) |
| 2 | 218,615 (19.8) | 104,442 (19.9) | 114,173 (19.7) |
| 3 | 213,759 (19.4) | 101,724 (19.4) | 112,035 (19.4) |
| 4 | 218,938 (19.8) | 104,489 (19.9) | 114,449 (19.8) |
| 5 (most wealthy) | 222,792 (20.2) | 105,696 (20.2) | 117,096 (20.2) |
| Missing (%) | 0.0 | 0.0 | 0.0 |
| Urban residency, n (%) | 360,250 (32.6) | 170,339 (32.5) | 189,911 (32.8) |
| Missing (%) | 0.0 | 0.0 | 0.0 |
| Currently married, n (%) | 839,697 (76.2) | 394,269 (75.3) | 445,428 (77.0) |
| Missing (%) | 0.1 | 0.2 | 0.1 |

[a]Data are not weighted to adjust for the survey design.
[b]Source data are provided as a Source Data file.
n number; % percentage

**Table 2 Cluster characteristics[a].**

|  | Household | Community | District | State |
|---|---|---|---|---|
| Number of clusters | 515,689 | 17,841 | 561 | 32 |
| Mean cluster size (SD) | 3.4 (1.65) | 109.4 (97.7) | 2415 (973.2) | 57,427 (27961.7) |
| Median cluster size (IQR) | 3 (2) | 66 (89) | 2270 (1302) | 59,792 (54,836) |

[a]Source data are provided as a Source Data file.
*SD* standard deviation, *IQR* interquartile range.

**Table 3 Clustering of cardiovascular disease risk factors at the state, district, community, and household level in India[a].**

| Risk factor | ICC state (95% CI) | ICC district (95% CI) | ICC community (95% CI) | ICC household (95% CI) |
|---|---|---|---|---|
| Raised BG | 0.031 (0.016–0.049) | 0.034 (0.030–0.038) | 0.089 (0.087–0.091) | 0.142 (0.140–0.145) |
| Raised BP | 0.023 (0.012–0.036) | 0.034 (0.030–0.038) | 0.065 (0.063–0.067) | 0.104 (0.102–0.107) |
| Current smoker | 0.090 (0.048–0.140) | 0.063 (0.056–0.070) | 0.131 (0.128–0.134) | 0.095 (0.093–0.097) |
| Overweight | 0.045 (0.023–0.072) | 0.073 (0.065–0.081) | 0.134 (0.132–0.137) | 0.236 (0.234–0.239) |
| Obesity | 0.029 (0.015–0.046) | 0.039 (0.034–0.044) | 0.099 (0.096–0.101) | 0.165 (0.163–0.167) |

[a]Source data are provided as a Source Data file.
*ICC* intracluster correlation coefficient, *CI* confidence interval, *BG* blood glucose, *BP* blood pressure

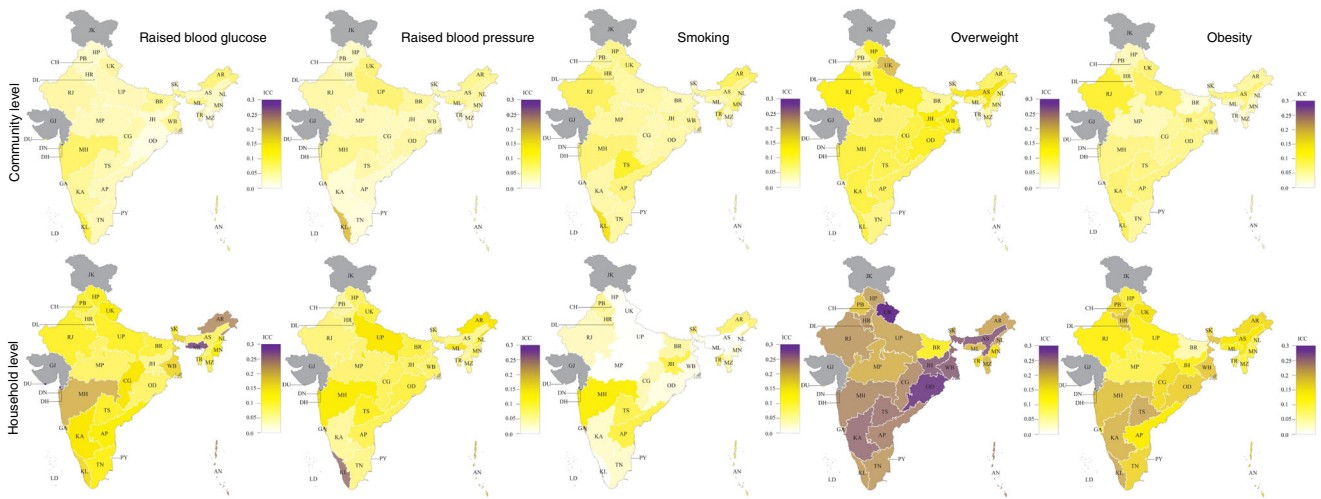

**Fig. 1 Intracluster correlation coefficients at the household and community level by state.** *AN* Andaman and Nicobar Islands, *AP* Andhra Pradesh, *AR* Arunachal Pradesh, *AS* Assam, *BR* Bihar, *CG* Chhattisgarh, *CH* Chandigarh, *DD* Daman and Diu, *DL* Delhi, *DN* Dadra and Nagar Haveli, *GA* Goa, *GJ* Gujarat, *HR* Haryana, *HP* Himachal Pradesh, *JH* Jharkhand, *JK* Jammu and Kashmir, *KA* Karnataka, *KL* Kerala, *LD* Lakshadweep, *MP* Madhya Pradesh, *MH* Maharashtra, *MN* Manipur, *ML* Meghalaya, *MZ* Mizoram, *NL* Nagaland, *OD* Odisha (Orissa), *PB* Punjab, *PY* Puducherry, *RJ* Rajasthan, *SK* Sikkim, *TN* Tamil Nadu, *TS* Telangana State, *TR* Tripura, *UP* Uttar Pradesh, *UK* Uttarakhand (Uttaranchal), *WB* West Bengal. Source data are provided as a Source Data file.

decreased in size from the state to the household level. We did not observe any clear patterns of variation in these ICCs when computing ICCs separately for each household wealth quintile (Supplementary Table 11).

**Clustering at the household and community level by state.** There was a large degree of variation in the ICCs for each CVD risk factor between states (Fig. 1 and Supplementary Tables 12 and 13), whereby we found the highest ICCs within states for overweight at the household level and the lowest for smoking at the household level. On an absolute scale, the variation in the ICC between states was smallest for smoking. For a given CVD risk factor, the pattern of variation in the ICC between states was generally similar for households as for communities.

In rural areas, household-level ICCs within districts were positively associated with a district's median household wealth index for all CVD risk factors except raised blood pressure (Fig. 2). These associations were generally not present in urban areas. Previous articles have shown a link between the prevalence and the ICC for binary outcomes[17–20]. To examine this tendency we performed a regression of the CVD risk factor prevalence onto ICC in the appendix (Supplementary Fig. 1). We generally found no correlation between CVD risk factor prevalence and ICC, with the exception of a positive correlation between state-wise obesity prevalence and household-level ICC (Supplementary Fig. 1). We also did not observe any clear correlations between household- and community-level ICCs within states and states' gross domestic product (GDP) per capita (Supplementary Fig. 2).

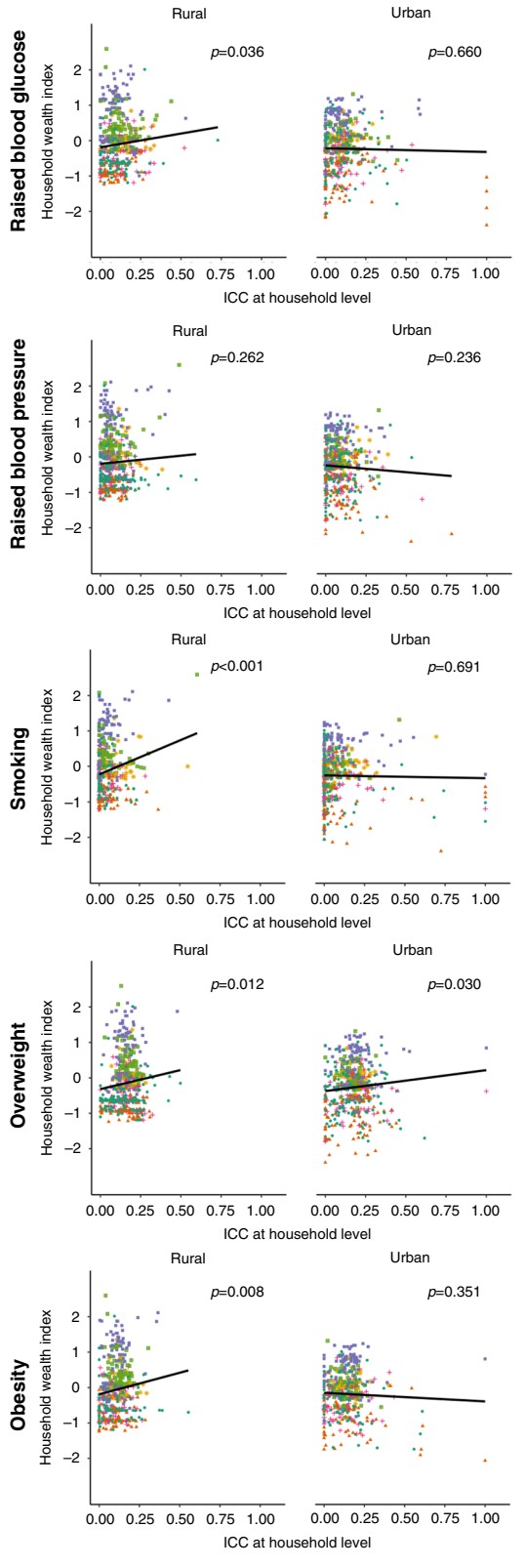

**Fig. 2 Intracluster correlation coefficients in relation to household wealth index by district, stratified by residency.** The black line is an ordinary least squares regression of district-level household wealth index onto household-level ICC with each district having the same weight. The *p* value (derived from a *t* test) refers to the regression coefficient for this black line. Colors designate the different zones in India as per the allocation of the Zonal Councils of the Government of India: green circle = Central, orange triangle = East, purple square = North, pink cross = Northeast, green square = South, yellow star = West[52]. For the calculation of the ICCs we included districts with ≥50 participants and ≥20 participants with the respective CVD risk factor. All 561 districts except for one had ≥50 participants. For rural areas, 53 districts had <20 individuals with raised blood glucose, all had ≥ 20 participants with raised blood pressure, seven districts had <20 participants who were currently smoking, two districts had <20 participants with overweight, and 74 had <20 participants with obesity. For urban areas, 44 districts had <20 individuals with raised blood glucose, all had ≥20 participants with raised blood pressure, five districts had <20 participants who were currently smoking, one district had <20 participants with overweight, and 68 had <20 participants with obesity. Source data are provided as a Source Data file.

states). Except for smoking, the level of clustering was highest for households, followed by communities, districts, and then states. The ICCs were particularly high for the clustering of overweight at the household level. We also identified a wide degree of variation of clustering at the household and community level between states. In addition, in rural areas, CVD risk factors tended to cluster more strongly at the household level in wealthier districts than in poorer ones. Although previous studies in India have shown strong concordance of chronic conditions between household members and by different family relationships[21], this is, to our knowledge, the first study to examine the variation of clustering of CVD risk factors between socio-geographic levels and states in India.

Our findings can inform design decisions in cluster-randomized trials and household surveys. In cluster-randomized trials (e.g., those that randomize households or villages to different study arms but observe and analyze the outcome at the individual level) and household surveys the sample size is highly dependent on the expected ICC of the outcome variable[22]. Unfortunately, ICC estimates for a specific outcome variable for a chosen socio-geographic level and a given study setting are usually not available, such that researchers can only guess what a reasonable ICC range might be. By providing ICC values for each of the CVD risk factors for each state and each socio-geographic level in India, this study, thus, provides urgently needed information for the design of cluster-randomized trials and household surveys on CVD risk factors in India. We have made our data set and code publicly available such that researchers can modify our ICC calculations as needed (e.g., for different age ranges and outcome definitions)[23].

In addition to their usefulness for the design of trials and household surveys, our findings can inform the targeting of policies and programming for the reduction of CVD risk factors. From a program implementation perspective, based on our findings of a high degree of clustering of overweight at the household level, health promotion interventions aimed at reducing overweight might be especially effective when targeted at this level instead of targeting higher levels or the general population. Conversely, from a research perspective, the greater the ICC is in a cluster-randomized trial, the more participants are required to reach a given level of statistical power. The same is not true for smoking, however, which displayed little clustering at the household level. A likely reason is that smoking prevalence in India seems to be strongly driven by gender- rather than family-

## Discussion

Using population-based data on over one million adults across India, this study identified a wide degree of variation of the level of clustering of CVD risk factors between risk factors and type of socio-geographic level (households, communities, districts, or

related factors, given that men are far more likely to smoke than woman. In addition, we have shown in which states targeting particular socio-geographic levels would be most efficient. For instance, the level of clustering of overweight at the household level was especially high in Uttarakhand and Odisha, whereas it was far lower in Bihar and Manipur. More generally, our results show that the ICC tended to increase as the socio-geographic unit decreased in size. This is consistent with previous research from different surveys and settings[24–29]. One reason for this finding in our study may be that diet- and lifestyle-related factors, which in turn influence the CVD risk factors that we examined, tend to be shared within communities and especially households (which may, for example, share meals).

Our study has several limitations. First, the AHS did not report fasting status of participants, which impacts the BG cutoff used to define raised BG. Although we assumed all participants were fasted in our primary analysis, we show all ICC results for raised BG when assuming AHS participants to be unfasted in the appendix (Supplementary Tables 14–16). Second, 31.9% of participants had a missing value for at least one of the CVD risk factors examined in this study. Participants with missing data may have had a different prevalence of these risk factors than those for whom we had complete data. This may have biased our ICC estimates if individuals with a systematically different prevalence of these risk factors (but for whom we had missing observations) were more or less likely to cluster within the socio-geographic unit in question than those for whom we had complete data. Third, for participants in the AHS, sociodemographic information had to be matched to physical measurements, which were contained in a separate data set provided by the Ministry of Health & Family Welfare. Given the lack of a unique identifier in both datasets, we used a matching technique described in Text S1. Although the sociodemographic characteristics of those who were matched were similar to those who were not matched, it is possible that those who could be matched are not a random subset of all AHS participants. Fourth, owing to violent conflicts and unavailable data in the public domain, the AHS and DLHS-4 contain no data on the states of Jammu and Kashmir and Gujarat, as well as the UTs of Dadra and Nagar Haveli and Lakshadweep. However, these states and UTs constituted only 6% of India's population at the time of the 2011 India census[30]. Fifth, our findings cannot necessarily be extrapolated to the current year because the degree to which CVD risk factors are affected by the socio-geographic environment may have changed since the time of data collection. Similarly, researchers and policymakers should be cautious with extrapolating our findings to populations other than India, as the nature of the socio-geographic units that we examined and the degree to which they are associated with different CVD risk factors is likely to be context-specific and thus to vary across countries.

The level of clustering of CVD risk factors varies widely between risk factors, socio-geographic levels, and states in India. By detailing this variation in a large population-based data set and for each of five CVD risk factors, this analysis provides critical information both for the design of cluster-randomized trials and household surveys, as well as targeting of relevant policies and prevention programming in India.

## Methods

**Data sources**. We used data from two population-based household surveys in India: the fourth District-Level-Household Survey (DLHS-4) and the second update of the Annual Health Survey (AHS). We pooled both datasets because they were conducted at the same time, used a nearly identical questionnaire, and covered mutually exclusive areas of the country, jointly yielding a nationally representative sample of adults in India.

The AHS was carried out from 2012 to 2013 and covered nine states (Assam, Bihar, Chhattisgarh, Jharkhand, Madhya Pradesh, Odisha, Rajasthan, Uttar

Pradesh, and Uttarakhand), which were chosen because they had the highest rate of child and infant mortality in 2010. The DLHS-4 was carried out between 2012 and 2014, and covered all remaining states (except Gujarat for which data was not available in the public domain, and Jammu and Kashmir for which data was not collected due to violent conflicts) and five of seven Union Territories (UTs) (all except Dadra and Nagar Haveli, and Lakshadweep). Both surveys sampled non-pregnant adults aged 18 years and older.

The DLHS-4 and AHS surveys employed two-stage cluster random sampling, stratified by rural-urban location. The primary sampling units (PSUs)—villages in rural areas and census enumeration blocks (AHS) or urban frame survey blocks (DLHS-4) in urban areas—were selected with probability proportional to population size. PSUs are henceforth referred to as communities in this manuscript. The secondary sampling units were households, which were selected via systematic random sampling (i.e., sampling the first household randomly and then selecting every xth household). The household head completed a questionnaire on sociodemographic information of all household members (regardless of their presence at the interviewer's visit) and all non-pregnant adult household members (aged ≥ 18) received height, weight, blood glucose (BG), and blood pressure (BP) measurements. Households were not revisited when eligible adults were not present for the interviewer's visit.

In an effort to ensure high data quality, both the DLHS-4 and AHS collected every tenth blood sample in duplicate (to then compare measurements taken for the same participants) and, in the AHS, 10% of households were revisited to administer part of the questionnaire a second time in order to detect problems with the questionnaire-based data collection[31,32].

**CVD risk factors**. Both the AHS and DLHS-4 measured BG in a capillary blood sample with a hand-held BG meter (SD CodeFree [SD Biosensor, Gyeonggi, Republic of Korea]). The glucose meter converted capillary blood measurements to a plasma-equivalent value by multiplying the reading by 1.11[33]. Both the AHS and DLHS-4 survey requested overnight fasting until the BG measurement but only the DLHS-4 recorded whether participants reported to have been adhering to this recommendation. In all, 58.4% of participants in the DLHS-4 self-reported to have been fasted at the time of the measurement. Although all estimates in this paper assume AHS participants to have been fasted at the time of the BG measurement, we also show results when assuming AHS participants to not have been fasted in the appendix. Raised BG was defined as a plasma-equivalent glucose >126 mg/dl if reporting (or assumed) to be fasted, or ≥200 mg/dl if reporting to be non-fasted[33,34].

In both surveys, BP was measured twice in the left upper arm with a minimum of three minutes in between measurements using the Rossmax AW150 (Rossmax Swiss GmbH, Bernick, Switzerland). We used the mean of these two measurements. Raised BP was defined as a mean systolic BP ≥ 140 mmHg or mean diastolic BP ≥ 90 mmHg[35].

Weight was measured in both surveys with a digital scale, and height using a wall-mounted statute meter. Body Mass Index (BMI) was calculated as weight in kilograms divided by the square of height in meters (kg m$^{-2}$). Using the World Health Organization (WHO) classification for Asian populations[36–38], we defined overweight as a BMI > 23.0 kgm$^{-2}$ and obesity as a BMI > 27.5 kgm$^{-2}$.

Current tobacco smoking status was ascertained through self-report in the questionnaires. Participants who reported to smoke at least once every day or to be occasional smokers were classified as currently smoking.

For the use in future cluster-randomized trials, we have additionally run all analyses for the continuous variables of BMI, BG, and BP (Supplementary Tables 17–19, Supplementary Figs. 3 and 4).

**Statistical analysis**. Sociodemographic data in the AHS were matched to data on individuals' BG, BP, and height and weight measurements (contained in a separate dataset) as described in Supplementary Methods. We computed ICCs as a measure of clustering at different socio-geographic levels. The socio-geographic levels considered were the household, community, district, and state. The ICC indicates the proportion of the variation in the outcome that is due to variation between—rather than within—clusters (e.g., households). An ICC value of zero thus indicates that the socio-geographic clusters do not account for any of the variation in the CVD risk factor among respondents. The ICCs were calculated using a linear model with a random intercept for the socio-geographic level:

$$Y_{ij} = \beta_0 + u_{0j} + \varepsilon_{ij}, \qquad (1)$$

where $Y_{ij}$ is a binary variable indicating whether individual $i$ in cluster $j$ has the CVD risk factor in question, $\beta_0$ is the overall intercept, and $u_{0j}$ is the random intercept for each cluster. The variance ($\sigma$) of $u_{0j}$ is, thus, the between-cluster variance, and $\sigma(\varepsilon_{ij})$ is the within-cluster variance[39]. Hence, the ICC can be calculated as the between-cluster variance divided by the total variance[39–41]:

$$\text{ICC} = \frac{\sigma(u_{oj})}{\sigma(u_{oj}) + \sigma(\varepsilon_{ij})} \qquad (2)$$

95% confidence intervals (CIs) for the ICC were calculated using parametric bootstrapping for mixed models with 500 repetitions[42,43]. We did not adjust the regressions for individual-level characteristics because we were interested in determining the degree to which CVD risk factors cluster at a certain socio-

geographic level—which in turn determines the efficiency of targeting that level with relevant interventions—regardless of the degree to which the between-cluster variance can be explained by differences between individuals across clusters. Calculating unadjusted ICCs is also of most relevance to informing power calculations for cluster-randomized trials, which typically use unadjusted analyses as their primary approach.

We have chosen to present an ICC from a linear rather than a logistic regression model because the aim of this study is to inform the design effect in future cluster-randomized trials or surveys that use cluster sampling, and to examine the correlation in the outcomes between participants[44]. When calculating ICCs, we only included a random effect for the socio-geographic level that we examined (e.g., we did not additionally include random effects for household or community when determining the district-level ICC). When calculating ICCs for different states, by wealth status, and by rural-urban residency, we fitted the regression model to only the subgroup of interest.

To examine how the degree of clustering at each socio-geographic level varied within India, we disaggregated the ICC values by state and wealth quintile. A given cluster was assigned to a wealth quintile based on a continuous household wealth index. To compute the household wealth index, we ran a principal component analysis on the answers to household ownership of 12 assets (radio, TV, computer, phone, refrigerator, bike, scooter, car, washing machine, sewing machine, house, and land) and five housing characteristics (water supply, type of toilet and whether it is shared, cooking fuel, housing material, and source of lighting) from which we then extracted the first component as per the methodology developed by Filmer and Pritchett[45,46]. The household wealth index was calculated separately for rural and urban areas. Households were then divided—again separately for rural and urban areas—into five quintiles based on the distribution of the continuous household wealth index in the combined DLHS-4/AHS data set. We calculated the median of the continuous household wealth index in a community or district based on which we then assigned communities and districts to a wealth quintile.

In addition, we plotted the district-level median of the continuous household wealth index variable (separately for rural and urban areas) against household-level ICC values to ascertain whether the wealth of a district explained some of the variation we observed in the household-level ICC values between districts. Similarly, we plotted the state-level prevalence of each CVD risk factor and GDP per capita against the household and community-level ICC values for each CVD risk factor to examine to what degree these variables explained the variation in the ICC values for community and household between states.

This was a complete case analysis. Statistical analyses were performed in R version 3.4.3[47]. Fig. 1 and Supplementary Fig. 3 have been created using Adobe Illustrator CC 2019, using a template of an Indian map from Wikimedia Commons[48,49].

**Reporting summary**. Further information on research design is available in the Nature Research Reporting Summary linked to this article.

## Data availability
The conclusions of this article are based on publicly available data sets. The source data are provided with this paper, the cleaned and merged dataset is available at https://doi.org/10.7910/DVN/NLU7HI[50,51]. Source data are provided with this paper.

## Code availability
The analysis code and the merged data set is available on the Harvard Dataverse (https://doi.org/10.7910/DVN/NLU7HI)[23].

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

## Acknowledgements

This project has received funding from the European Research Council (ERC) under the European Union's Horizon 2020 research and innovation program (grant agreement no. 850896). P.G. was supported by the National Center for Advancing Translational Sciences of the National Institutes of Health under Award Number KL2TR003143. J.W. was supported by the Alexander von Humboldt Foundation. The funders had no role in study design, analysis, or the decision to submit for publication.

## Author contributions

P.G., A.B., A.A., S.V., J.W., and T.B. conceptualized the design and analysis. A.A., A.B., and P.G. acquired the data. A.B. and P.G. analyzed the data and created the manuscript draft. A.B. created the figures and tables. P.G. obtained funding. All authors were closely involved in the interpretation of the data and revision of the manuscript. Each author has read and approved the final manuscript.

## Funding

## Competing interests

The authors declare no competing interests. The study received a determination of "Not Human Subjects Research" by the institutional review board of the Harvard T.H. Chan School of Public Health on 9 May 2018.
