## [Peer Review File · Nature Communications]

REVIEWER COMMENTS

Reviewer #2 (Remarks to the Author):

Thank you for submitting this manuscript. I found it very interesting to read through.

I have a few minor comments, which I have listed below:

I appreciate that BMI is often categorised into overweight and obese for the purpose of targeted interventions or describing risk. I wonder though whether you considered estimating the ICC for BMI in its raw continuous form? This equally applies to blood glucose and blood pressure. One of the aims of this piece of work is to describe the correlation and provide ICCs for future cluster randomised trials. An ICC for a binary outcome is only appropriate in a setting in which the prevalence of the outcome is the same as the one in which the ICC was calculated. By dichotomising BMI, blood pressure and blood glucose, you may be limiting the applicability of the results. However, this hasn't been mentioned in the methods or discussion.

You have described the characteristics of the individual participants. I wonder whether it would be possible to summarise the characteristics of the different clustering variables in some format. For example, describing the number of clusters, the average cluster size and a measure of variation in size. When there are few clusters (such as states here), the ICC estimate may need to be treated with caution.

Your results showed that generally the ICC increased as the socio-geographic unit decreased. Do you know whether this result is consistent with previous work? Do you have any thoughts on why this is the case?

There are a number of articles published showing a link between the prevalence and the ICC for binary outcomes – so that lower prevalence leads to a smaller ICC. It would be useful for you to comment on whether you have found similar, particularly given the number of subgroups you have investigated.

On page 11, you mention that health promotions aiming to reduce obesity may be more efficient if targeting households because of a higher degree of clustering. In a cluster trial, the greater the ICC, the more participants that are required. As such, it would be useful if you could elaborate on why

greater clustering would lead to a more efficient trial. Or at least what is meant by efficiency in this context.

There are two types of ICCs that can be calculated for binary outcomes. One type of ICC for binary outcomes – which we may call the “natural” ICC reflects the correlation between observations within a cluster, and is calculated as you mention, as a ratio of the between cluster variance component and the total variance, assuming a random effects linear model has been fitted. Another type of ICCs for binary outcomes – which we may call the “latent” ICC – can be calculated after fitting a random effects logistic regression model. Previous articles reporting ICCs have often not clarified which of these two different ICCs have been reported. Since you are interested in the correlation estimates for a future trial, and to describe the correlation between participants, the ICC you are reporting is most appropriate. However, I feel it would be useful to acknowledge the rationale behind choosing a linear model to estimate the ICCs, as some people may find it confusing that you have fitted linear models to binary outcomes.

When considering the higher hierarchical levels of clustering, such as state, do you still include random effects components for the lower levels of clustering, such as household?

You have calculated the ICCs separately for different states, wealth status, and urban/rural status. I assume this was done by fitting a model to just the subgroup of interest, as this is not explicitly stated.

Figure S1 is not particularly clear and easy to read, though this may be because of how each state is indicated. I am not sure if anything is lost if the states aren't indicated on the figure.

James Martin

Reviewer #3 (Remarks to the Author):

This is an analysis of cardiovascular disease (CVD) risk factors clustering within different socio-geographic levels, performed in India

They used nationally representative household survey data from India. What was the observer reproducibility for these data?

Also, out of 1,618,359 adults in our final dataset, 32% had missing values

This raises questions about the comprehensiveness of the analysis

The ICCs ranged from 0.019 for raised blood glucose at the community level to 0.236 for overweight at the household level. Except for smoking, the level of clustering was generally highest for households, followed by communities, districts, and then states.

More economically developed districts had a higher household ICC in rural areas. No surprise here.

The level of clustering of CVD risk factors varies widely between risk factors, socio- geographic levels, and states in India. Likely to vary in different parts of the world too!

What about changes in risk factors over time, with incident comorbidities and ageing, as well as drug therapies which may mitigate risk factors

This maybe useful for designing cluster RCTs in India but the generalisability needs to be considered.

In terms of practical clinical management, not sure about the applicability to everyday practice

Point-by-point response to the reviewer's comments

Reviewer #2

1. I appreciate that BMI is often categorised into overweight and obese for the purpose of targeted interventions or describing risk. I wonder though whether you considered estimating the ICC for BMI in its raw continuous form? This equally applies to blood glucose and blood pressure. One of the aims of this piece of work is to describe the correlation and provide ICCs for future cluster randomised trials. An ICC for a binary outcome is only appropriate in a setting in which the prevalence of the outcome is the same as the one in which the ICC was calculated. By dichotomising BMI, blood pressure and blood glucose, you may be limiting the applicability of the results. However, this hasn't been mentioned in the methods or discussion.

Authors' response: We thank the reviewer for this thoughtful comment, which we have now addressed in the following two ways. First, we have substantially expanded our analyses to further increase the applicability of our results. Specifically, we have now run all analyses for the continuous variables of BMI, blood glucose and blood pressure (in addition to dichotomising BMI, blood pressure and blood glucose), which is the recommended strategy by the reviewer (shown in Figures S3-S4 and Tables S7-S9). We find that the ICCs for blood glucose are much higher, while they are similar for blood pressure and differing without a clear pattern for BMI. For Figure 1, the state-wise ICC pattern remained similar. However, for continuous variables no association with the district's median wealth index could be found. We have further discussed these analyses in the Methods and Results sections, where we write:

“For the use in future cluster-randomized trials, we have additionally run all analyses for the continuous variables of BMI, blood glucose and blood pressure (Tables S7-S9, Figure S3-S4).” (p.12, Methods section, revised manuscript)

“Table S7. Clustering of BMI, BG and BP as continuous variables at the state, district, community, and household level in India” (p.17, revised supplementary appendix)

“Table S8: Intraclass correlation coefficients for BMI, BG and BP as continuous variables at the household level by state” (p.17, revised supplementary appendix)

“Table S9: Intraclass correlation coefficients for BMI, BG and BP as continuous variables at the community level by state” (p.18, revised supplementary appendix)

“Figure S3: Intracluster correlation coefficients for BMI, BG and BP as continuous variables at the household and community level by state” (p.25, revised supplementary appendix)

¹ Abbr.: AN, Andaman and Nicobar Islands; AP, Andhra Pradesh; AR, Arunachal Pradesh; AS, Assam; BR, Bihar; CG, Chhattisgarh; CH, Chandigarh; DL, Delhi; DN, Dadra and Nagar Haveli; GA, Goa; HR, Haryana; HP, Himachal Pradesh; JH, Jharkhand; KA, Karnataka; KL, Kerala; ; MP, Madhya Pradesh; MH, Maharashtra; MN, Manipur; ML, Meghalaya; MZ, Mizoram; NL, Nagaland; OD, Odisha (Orissa); PB, Punjab; PY, Puducherry; RJ, Rajasthan; SK, Sikkim; TN, Tamil Nadu; TS, Telangana State; TR, Tripura; UP, Uttar Pradesh; UK, Uttarakhand (Uttaranchal); WB, West Bengal.

“Figure S4: Intraclass correlation coefficients for BMI, BG and BP as continuous variables in relation to household wealth index by district, stratified by residency” (p.26, revised supplementary appendix)

¹ The black line is an ordinary least squares regression of district-level household wealth index onto household-level ICC with each district having the same weight. The p-value refers to the regression coefficient for this black line.

² Colors designate the different zones in India as per the allocation of the Zonal Councils of the Government of India. ¹

³ For the calculation of the ICCs we included districts with ≥ 50 participants and when separated by residency only districts with ≥ 20 participants. All 561 districts except for one had ≥ 50 participants. Then, for rural areas, 9 districts had ≤ 20 participants. For urban areas, 16 districts had ≤ 20 participants.

Second, we have now provided additional discussion on the applicability and generalizability of our findings more broadly. Specifically, we addressed the fact that our ICCs are targeted to the Indian population at the time of the survey and might not necessarily be extrapolatable to other countries or the current time.

“Fifth, our findings cannot necessarily be extrapolated to the current year because the degree to which CVD risk factors are affected by the socio-geographic environment may have changed since the time of data collection. Similarly, researchers and policymakers should be cautious with extrapolating our findings to populations other than India, since the nature of the socio-geographic units that we examined and the degree to which they are associated with different CVD risk factors is likely to be context-specific and thus to vary across countries.” (p.9, Discussion section, revised manuscript)

2. You have described the characteristics of the individual participants. I wonder whether it would be possible to summarise the characteristics of the different clustering variables in some format. For example, describing the number of clusters, the average cluster size and a measure of variation in size. When there are few clusters (such as states here), the ICC estimate may need to be treated with caution.

Authors’ response: We thank the reviewer for this excellent suggestion. We have now added a new table to the manuscript, which includes i) the number of clusters for each level, ii) the mean cluster size for each level with standard deviation, as well as iii) the median cluster size with interquartile range. We show results for the pooled sample (Table 2) and separately for each of the states and union territories in India (Table S10).

We find that the mean cluster size for households was 3.44, the mean community cluster size was 109.4 and the mean district cluster size was 2,415. When calculating the median cluster sizes, results were similar with a median of 3 on the household level, 66 on the community level, 2,270 on the district level and 59,792 on the state level with high interquartile ranges. Household-level cluster sizes were similar across states. On the community level, cluster sizes range from 37.6 to 237.3 (mean) or 37 to 221 (median), while district sizes range from 1190.8 to 4408.8 (mean) or 996 to 4690 (median).

“The mean cluster size of the 515,689 included households was 3.4 participants; the median cluster size was 3 participants (Table 2). In the 17,841 communities, a cluster consisted on average of 109.4 participants, with the median cluster size being 66 participants. On the district level, our analysis included 561 districts with a mean and median size of 2,415 and 2,270 participants, respectively. A state consisted of a mean and median of 57,427 and 59,792 participants, respectively. Cluster characteristics computed separately for each state can be found in the appendix (Table S10).” (p.5, Results section, revised manuscript)

“Table 2: Cluster characteristics” (p.21, Tables section, revised manuscript)

Table 2: Cluster characteristics

	Household	Community	District	State
Number of clusters	515,689	17,841	561	32
Mean cluster size (SD)	3.4 (1.65)	109.4 (97.7)	2,415(973.2)	57,427 (27961.7)
Median cluster size (IQR)	3 (2)	66 (IQR=89)	2,270 (IQR=1302)	59,792 (54,836)

Abbr. : SD=standard deviation, IQR=interquartile range

“Table S10: Cluster characteristics by state” (p.19, revised supplementary appendix)

3. Your results showed that generally the ICC increased as the socio-geographic unit decreased. Do you know whether this result is consistent with previous work? Do you have any thoughts on why this is the case?

Authors’ response: We thank the Reviewer for this comment. We have now further expanded the discussion of our results in light of prior evidence. The wide majority of articles confirms our findings that there is an inverse relation between ICC and cluster size.

“More generally, our results show that the ICC tended to increase as the socio-geographic unit decreased in size. This is consistent with previous research from different surveys and settings.²⁵⁻³⁰ One reason for this finding in our study may be that diet- and lifestyle-related factors, which in turn influence the CVD risk factors that we examined, tend to be shared within communities and especially households (which may, for example, share meals).” (p.8, Discussion section, revised manuscript)

References:

Campbell, M. K., Fayers, P. M. & Grimshaw, J. M. Determinants of the intracluster correlation coefficient in cluster randomized trials: the case of implementation research. *Clinical Trials* **2**, 99-107, doi:10.1191/1740774505cn071oa (2005).

Donner, A. An Empirical Study of Cluster Randomization. *International journal of epidemiology* **11** (1982).

Gulliford, M. C., Ukoumunne, O. C. & Chinn, S. Components of variance and intraclass correlations for the design of community-based surveys and intervention studies: data from the Health Survey for England 1994. *Am J Epidemiol* **149**, 876-883, doi:10.1093/oxfordjournals.aje.a009904 (1999).

Martinson, B. C., Murray, D. M., Jeffery, R. W. & Hennrikus, D. J. Intraclass correlation for measures from a worksite health promotion study: estimates, correlates, and applications.

American journal of health promotion : AJHP **13**, 347-357, doi:10.4278/0890-1171-13.6.347 (1999).

Pagel, C. *et al.* Intraclass correlation coefficients and coefficients of variation for perinatal outcomes from five cluster-randomised controlled trials in low and middle-income countries: results and methodological implications. *Trials* **12**, 151, doi:10.1186/1745-6215-12-151 (2011).

Smith, H. F. An empirical law describing heterogeneity in the yields of agricultural crops. *The Journal of Agricultural Science* **28**, 1-23, doi:10.1017/S0021859600050516 (1938).

4. There are a number of articles published showing a link between the prevalence and the ICC for binary outcomes – so that lower prevalence leads to a smaller ICC. It would be useful for you to comment on whether you have found similar, particularly given the number of subgroups you have investigated.

Authors' response: We thank the reviewer for bringing these articles to our attention, which we now discuss and cite in our revised manuscript. We also assessed the relationship between prevalence of CVD risk factors and ICC. To do so, we ran ordinary least squares regression models, regressing CVD risk factor state-prevalence onto household-level ICC (Figure S1, panel A), and CVD risk factor state-prevalence onto community-level ICC (Figure S1, panel B). We generally did not find a correlation between CVD risk factor state-prevalence and ICC, except for a positive relation between obesity prevalence and household-level ICC.

“Previous articles have shown an association between the prevalence and the ICC for binary outcomes.¹⁷⁻²⁰ To examine this in our data, we used ordinary least squares regression to regress CVD risk factor prevalence onto ICC (**Figure S1**). We generally found no correlation between CVD risk factor prevalence and ICC, with the exception of a positive correlation between state-level obesity prevalence and household-level ICC.” (p.6, Results section, revised manuscript)

References:

Gulliford, M. C. *et al.* Intraclass correlation coefficient and outcome prevalence are associated in clustered binary data. *Journal of Clinical Epidemiology* **58**, 246-251, doi:<https://doi.org/10.1016/j.jclinepi.2004.08.012> (2005).

Littenberg, B. & MacLean, C. D. Intra-cluster correlation coefficients in adults with diabetes in primary care practices: the Vermont Diabetes Information System field survey. *BMC Medical Research Methodology* **6**, 20, doi:10.1186/1471-2288-6-20 (2006).

Mickey, R. M. & Goodwin, G. D. The magnitude and variability of design effects for community intervention studies. *Am J Epidemiol* **137**, 9-18, doi:10.1093/oxfordjournals.aje.a116606 (1993).

Taljaard, M. *et al.* Intracluster correlation coefficients from the 2005 WHO Global Survey on Maternal and Perinatal Health: implications for implementation research. *Paediatric and perinatal epidemiology* **22**, 117-125, doi:10.1111/j.1365-3016.2007.00901.x (2008).

5. On page 11, you mention that health promotions aiming to reduce obesity may be more efficient if targeting households because of a higher degree of clustering. In a cluster trial, the greater the ICC, the more participants that are required. As such, it would be useful if you could elaborate on why greater clustering would lead to a more efficient trial. Or at least what is meant by efficiency in this context.

Authors' response: We thank the reviewer for this thoughtful input. We aimed to point out that for politicians and health program developers our findings of a high degree of overweight clustering at the household level are particularly useful, as these programs will be more successful when targeted at the household level instead of the general population or state level. We have now further clarified this in the discussion section, where we write:

“From a program implementation perspective, based on our findings of a high degree of clustering of overweight at the household level, health promotion interventions aimed at reducing overweight might be especially effective when targeted at this level instead of targeting higher levels or the general population. Conversely, from a research perspective, the greater the ICC is in a cluster-randomized trial, the more participants are required to reach a given level of statistical power.” (p.7, Discussion section, revised manuscript)

6. There are two types of ICCs that can be calculated for binary outcomes. One type of ICC for binary outcomes – which we may call the “natural” ICC reflects the correlation between observations within a cluster, and is calculated as you mention, as a ratio of the between cluster variance component and the total variance, assuming a random effects linear model has been fitted. Another type of ICCs for binary outcomes – which we may call the “latent” ICC – can be calculated after fitting a random effects logistic regression model. Previous articles reporting ICCs have often not clarified which of these two different ICCs have been reported. Since you are interested in the correlation estimates for a future trial, and to describe the correlation between participants, the ICC you are reporting is most appropriate. However, I feel it would be useful to acknowledge the rationale behind choosing a linear model to estimate the ICCs, as some people may find it confusing that you have fitted linear models to binary outcomes.”

Authors' response: We thank the reviewer for this important comment. We completely agree that it is important to acknowledge more clearly our rationale for choosing a linear model on binary outcomes to estimate ICC values. We have now further clarified this. Specifically, we have underscored that we aim to inform the design effect for sample size calculations in future trials and cluster surveys, for which sample size calculations should not use the “latent” ICC.

“We have chosen to present an ICC from a linear rather than a logistic regression model because the aim of this study is to inform the design effect in future cluster-randomized trials or surveys that use cluster sampling, and to examine the correlation in the outcomes between participants.⁴⁴” (p.13, Methods section, revised manuscript)

References :

Martin, J. *et al.* Intra-cluster and inter-period correlation coefficients for cross-sectional cluster randomised controlled trials for type-2 diabetes in UK primary care. *Trials* **17**, 402-402, doi:10.1186/s13063-016-1532-9 (2016).

7. When considering the higher hierarchical levels of clustering, such as state, do you still include random effects components for the lower levels of clustering, such as household? You have calculated the ICCs separately for different states, wealth status, and urban/rural status. I assume this was done by fitting a model to just the subgroup of interest, as this is not explicitly stated.

Authors’ response: Indeed, when calculating ICCs separately for different subgroups, the model exclusively included random effect components for the corresponding level. We have now clarified this in the Methods section of the revised manuscript, where we write:

“When calculating ICCs, we only included a random effect for the socio-geographic level that we examined (e.g., we did not additionally include random effects for household or community when determining the district-level ICC). When calculating ICCs for different states, by wealth status, and by rural-urban residency, we fitted the regression model to only the subgroup of interest.” (p.13, Methods section, revised manuscript)

8. Figure S1 is not particularly clear and easy to read, though this may be because of how each state is indicated. I am not sure if anything is lost if the states aren’t indicated on the figure.

Authors’ response: We have now redrawn Figure S1 without names of states and union territories to further improve readability, as recommended.

“Figure S1: Intracluster correlation coefficients in relation to CVD risk factor prevalence by state” (p. 21, revised supplementary appendix)

Reviewer #3

1.They used nationally representative household survey data from India. What was the observer reproducibility for these data?

Authors' response: We thank the reviewer for this comment. Unfortunately, the survey implementer (the International Institute for Population Sciences) did not make any data available on inter- nor intra-observer reproducibility. However, both the DLHS-4 and AHS took a number of measures in an effort to ensure high data quality. Specifically, every 10th blood sample was collected in duplicate in both surveys in order to detect problems with the blood-based measurements. Additionally, 10% of sampled households were visited again by medical consultants in the AHS and part of the questionnaire was re-administered to the same participants in order to detect problems with the questionnaire-based data collection.

“In an effort to ensure high data quality, both the DLHS-4 and AHS collected every tenth blood sample in duplicate (to then compare measurements taken for the same participants) and, in the AHS, ten percent of households were revisited to administer part of the questionnaire a second time in order to detect problems with the questionnaire-based data collection.^{31,32}” (p.11, Methods section, revised manuscript)

References:

Ministry of Health and Family Welfare.District level household & facility survey (DLHS-4)-Field Operational Manual for Health Investigators /Supervisors. (Government of India, Mumbai, 2012-2013).

Office of the Registrar General and Census Commissioner. Annual Health Survey Report - A Report on Clinical, Anthropometric and Bio-Chemical Survey Part II. (Office of the Registrar General and Census Commissioner, India, New Delhi, India).

2.Also, out of 1,618,359 adults in our final dataset, 32% had missing values. This raises questions about the comprehensiveness of the analysis.

Authors' response: Conducting such a large household survey as the DLHS-4 and AHS is logistically complex, and we, thus, expected a certain degree of missing data. However, we do agree that the level of missingness is a potential source of bias in this study. One way of dealing with missing data is multiple imputation. We felt this approach would not be warranted in this data because we would primarily be using individuals' sociodemographic characteristics to predict their missing CVD risk factor levels. Sociodemographic characteristics in turn, however, would be expected to be highly correlated with the socio-geographic units for which we computed ICCs. The multiple imputation process may, thus, introduce more bias into our analysis rather than reducing bias. Nevertheless, we would be happy to conduct a multiple imputation if the reviewer feels this would be of advantage.

We have now emphasized in the discussion that the missing outcome data is a potential source of bias in our analysis:

“Second, 31.9% of participants had a missing value for at least one of the CVD risk factors examined in this study. Participants with missing data may have had a different prevalence of these risk factors than those for whom we had complete data. This may have biased our ICC estimates if individuals with a systematically different prevalence of these risk factors (but for whom we had missing observations) were more or less likely to cluster within the socio-geographic unit in question than those for whom we had complete data.” (p.8, Discussion section, revised manuscript)

3. More economically developed districts had a higher household ICC in rural areas. No surprise here.

Authors’ response: It is important to recognize that we are examining the ICC rather than prevalence. While we did expect that prevalence of CVD risk factors (other than smoking, which is more common among poorer population groups in India) would be higher in rural areas of more economically developed districts than in less economically developed districts, it was not evident to us prior to this analysis that CVD risk factors were more likely to cluster within households of more economically developed districts than they would within households of less economically developed districts.

4. The level of clustering of CVD risk factors varies widely between risk factors, socio-geographic levels, and states in India. Likely to vary in different parts of the world too! What about changes in risk factors over time, with incident comorbidities and ageing, as well as drug therapies which may mitigate risk factors. This maybe useful for designing cluster RCTs in India but the generalisability needs to be considered.

Authors’ response: We thank the reviewer for these comments. The aim of our analysis was never to be representative of the entire world. While we do recognize that there are important limits to the representativeness of our analysis, we strongly feel that representativeness is a key strength of our study. We have assembled a large dataset, which is representative of the entire Indian adult population (which accounts for over one sixth of the world’s population). In addition, we feel that we are exceptionally comprehensive in our analytical approach by examining all CVD risk factors recorded in the data and all socio-geographic levels that can be identified in the surveys.

Nevertheless, we do, of course, recognize that our study’s representativeness also has its limitations. While we do use the most recent nationally representative household survey data for the Indian adult population that is available, it is possible that some of the patterns that we observed in our study may have changed somewhat since the time of the survey. We have now substantially expanded our discussions of the generalizability of our findings across

subpopulations, indicators, settings, and time periods to highlight this limitation. Specifically, we underlined that our ICC values cannot necessarily be extrapolated to other countries or the current time and is most appropriate for the use in India or similar populations.

“Fifth, our findings cannot necessarily be extrapolated to the current year because the degree to which CVD risk factors are affected by the socio-geographic environment may have changed since the time of data collection. Similarly, researchers and policymakers should be cautious with extrapolating our findings to populations other than India, since the nature of the socio-geographic units that we examined and the degree to which they are associated with different CVD risk factors is likely to be context-specific and thus to vary across countries.” (p.9, Discussion section, revised manuscript)

5. In terms of practical clinical management, not sure about the applicability to everyday practice

Authors’ response: We would like to highlight that the motivation for our study was not to inform clinical practice. Instead, the utility of our study is twofold. First, we aimed to inform researchers who conduct cluster-randomized trials and/or household surveys (which usually employ cluster random sampling). The ICC has an important bearing on the sample size that is required for a given level of statistical power when conducting such a trial or survey.¹⁻³ Yet, because of lacking evidence, researchers usually have to resort to guessing the ICC or calculating the sample size for a range of “reasonable” ICCs. Second, our study aims to inform policymakers and program managers who plan screening or treatment interventions for CVD risk factors in India. As we have shown in several studies,⁴⁻⁷ the majority of individuals with important CVD risk factors, such as diabetes and hypertension, in low- and middle-income settings, including in India, are not diagnosed. This has led to calls to increase screening for these conditions through population-based approaches. It is, however, entirely unclear how such screening programs should be operationalized. Our analysis is a crucial contribution to these considerations because it can inform the degree to which an approach that targets certain socio-geographic units (e.g., certain districts, communities, or households) is a promising strategy as opposed to an approach that ignores these socio-geographic units (e.g., by aiming to screen everyone in a state who is above a certain age). We have now ensured that the motivation for our study is clear to readers from the very beginning by adding the following text to the introduction section:

“In this study, we used nationally representative household survey data from India to determine the intra-cluster correlation coefficients (ICCs) of five major CVD risk factors at each of four different socio-geographic levels (household, community, district, and state). In addition, we aimed to ascertain how the degree of clustering of CVD risk factors varies between states and by household wealth. The motivation for this study was not to inform individual-level clinical management. Instead, our objective was to provide critical information for sample size calculations of cluster-randomized trials and household surveys. In addition, understanding the degree to which important CVD risk factors tend to co-occur within these socio-geographic levels is crucial to inform the targeting of appropriate interventions. For instance, policymakers need to decide whether

to target a screening program for diabetes and hypertension at specific communities or types of households, or if they should instead disregard these socio-geographic units, such as by screening everyone above a certain age threshold.” (p.4, Introduction section, revised manuscript)

In addition, we have further added to the utility of our findings by conducting all analyses for BMI, blood glucose, and blood pressure as continuous as opposed to only as dichotomous variables. The ICCs resulting from these additional analyses will be useful to researchers who choose a continuous rather than a dichotomous outcome variable for their trial or household survey. We now refer to these additional analyses in the results section:

“For the use in future cluster-randomized trials, we have additionally run all analyses for the continuous variables of BMI, blood glucose, and blood pressure (Tables S7-S9, Figure S3-S4).” (p.12, Methods section, revised manuscript)

“Figure S3. Intraclass correlation coefficients for BMI, BG and BP as continuous variables at the household and community level by state” (p.25, revised supplementary appendix)

“Figure S4: Intraclass correlation coefficients for BMI, BG and BP as continuous variables in relation to household wealth index by district, stratified by residency^{1,2,3}” (p.26, revised supplementary appendix)

“Table S7. Clustering of BMI, BG and BP as continuous variables at the state, district, community, and household level in India” (p.17, revised supplementary appendix)

“Table S8: Intraclass correlation coefficients for BMI, BG and BP as continuous variables at the household level by state” (p.17, revised supplementary appendix)

“Table S9: Intraclass correlation coefficients for BMI, BG and BP as continuous variables at the community level by state” (p.18, revised supplementary appendix)

References:

- 1.Campbell MK, Mollison J, Grimshaw JM. Cluster trials in implementation research: estimation of intraclass correlation coefficients and sample size. *Statistics in Medicine* 2001; **20**(3): 391-9.
- 2.Goldstein H. *Multilevel Statistical Models*, 4th Edition; 2010.
- 3.Kerry SM, Bland JM. The intraclass correlation coefficient in cluster randomisation. *BMJ (Clinical research ed)* 1998; **316**(7142): 1455-60.
- 4.Prenissl J, Jaacks LM, Mohan V, et al. Variation in health system performance for managing diabetes among states in India: A cross-sectional study of individuals aged 15 to 49 years. *BMC Medicine* 2019; **17**(1): 92.

5.Prenissl J, Manne-Goehler J, Jaacks LM, et al. Hypertension screening, awareness, treatment, and control in India: A nationally representative cross-sectional study among individuals aged 15 to 49 years. *PLOS Medicine* 2019; **16**(5): e1002801.

6.Geldsetzer P, Manne-Goehler J, Theilmann M, et al. Diabetes and Hypertension in India: A Nationally Representative Study of 1.3 Million Adults. *JAMA internal medicine* 2018.

7.Manne-Goehler J, Geldsetzer P, Agoudavi K, et al. Health system performance for people with diabetes in 28 low- and middle-income countries: A cross-sectional study of nationally representative surveys. *PLoS Med* 2019; **16**(3): e1002751.

REVIEWERS' COMMENTS:

Reviewer #2 (Remarks to the Author):

Thank you very much for addressing the comments given previously. I think it clarifies some key issues, and will help readers to fully understand the methods, and the applicability of the results.

Reviewer #3 (Remarks to the Author):

No additional comments

Responses to my prior comments have simply to have added text to acknowledge the various Limitations